# Association Between Dietary Patterns and Cognitive Function in Midlife Adults: The Bogalusa Heart Study

**DOI:** 10.3390/nu17101636

**Published:** 2025-05-10

**Authors:** Kristen Ogarrio, Maria P. Santos, Ileana De Anda-Duran, Kaitlin S. Potts, Lydia A. Bazzano, Sylvia H. Ley

**Affiliations:** 1Department of Epidemiology, Celia Scott Weatherhead School of Public Health and Tropical Medicine, Tulane University, New Orleans, LA 70112, USA; kogarrio@tulane.edu (K.O.); msantos3@tulane.edu (M.P.S.); ideandaduran@tulane.edu (I.D.A.-D.); kspotts@bwh.harvard.edu (K.S.P.); lbazzano@tulane.edu (L.A.B.); 2Division of Sleep and Circadian Disorders, Department of Sleep Medicine, Brigham and Women’s Hospital, Harvard Medical School, Boston, MA 02115, USA

**Keywords:** diet quality, cognitive function, midlife adults, bogalusa heart study, mediterranean diet, healthy eating index, alternate healthy eating index

## Abstract

*Background*: Individual nutrients are associated with cognitive performance, but there is less evidence on the relationship between overall dietary patterns and cognitive performance in midlife. *Objective:* To examine the relation between dietary patterns and cognitive performance in midlife adults within the Bogalusa Heart Study (BHS). *Methods*: Cross-sectional data from the 2013–2016 cycle of the Bogalusa Heart Study, a life-course prospective cohort study, were used to generate diet quality scores, including the Alternate Healthy Eating Index 2010 (AHEI), the Healthy Eating Index 2015 (HEI), and the Alternate Mediterranean Dietary Pattern (aMed), based on food frequency questionnaires. Cognitive scores assessing attention and processing, episodic memory, and executive function were generated through validated cognitive tests. Generalized linear and logistic regression models were fit with adjustment for potential confounders. *Results*: Of 1053 participants included in the analysis, the mean age was 48.18 (SD = 5.22) years; 38.94% were male; and 31.14% identified as Black. Higher diet quality scores were associated with a higher global cognitive score (*P*-trend = 0.01 for AHEI, 0.02 for HEI, and 0.04 for aMed) after adjusting for age, sex, race, employment, education, smoking status, total energy intake, physical activity, BMI, diabetes, and hypertension. In dichotomous outcome analyses, higher AHEI and aMed scores remained inversely associated with low cognition when adjusting for similar covariates (*P*-trend = 0.03 for AHEI, 0.03 for aMed), but the association was attenuated to non-significance for HEI. In joint effect analysis between employment and dietary patterns (*P*-interaction = 0.28 for AHEI, 0.01 for HEI, and 0.11 for aMed), unemployment with a lower quality diet was associated with lower cognitive performance (*P*-trend = 0.02 for AHEI, 0.02 for HEI, and 0.01 for aMed). *Conclusions*: A high-quality diet is associated with optimal cognitive performance among midlife adults, and unemployment status may influence this association. These findings suggest that dietary interventions during midlife may represent a public health strategy to preserve cognitive function and reduce future neurodegenerative disease burden, especially in disadvantaged populations.

## 1. Introduction

Neurodegenerative diseases are among the leading causes of death in the United States, with Alzheimer’s disease ranked as the seventh most attributable cause of mortality [1]. Currently, approximately one in every nine American adults ages 65 and older has Alzheimer’s, with older Black Americans twice as likely to have it than older White Americans [2]. In 2024, an estimated 7 million Americans are affected by Alzheimer’s, and this number is expected to grow to 12.7 million by 2050 as the population of older adults increases [2]. While Alzheimer’s disease predominantly affects older adults, cognitive impairments such as declines in memory, executive function, and processing speed can emerge in midlife, a period when the brain begins to undergo physiological changes associated with neurodegenerative processes [3]. Therefore, intervening at this life stage may be important for protecting brain health, but evidence is scarce on how modifiable factors, such as diet quality, at midlife may impact cognitive performance.

Although midlife is considered a critical window for dietary intervention to help mitigate cognitive decline, evidence on the association between diet quality and cognitive function remains mixed [4,5,6,7,8,9,10,11,12,13]. Some studies suggest adherence to a high-quality diet or consumption of specific nutrients and food items, such as omega-3 fatty acids and whole grains [14], are linked to better cognitive function and slower cognitive decline in midlife [4,5,6,7]. Several studies have investigated the association between overall dietary patterns and cognitive function in adults [8,9,10,11,12,13]. Several large cohort studies have focused specifically on the Mediterranean diet, DASH, and the Alternate Healthy Eating Index (AHEI). In the U.S. Nurses’ Health Study, long-term adherence to these diets—each emphasizing fruits, vegetables, whole grains, nuts, and healthy fats—was associated with a lower risk of subjective cognitive decline [10]. The prospective U.K. Whitehall II study and the Swedish Malmö Diet and Cancer Study both reported that better midlife diet quality, particularly patterns aligned with AHEI components, was linked to a lower risk of developing dementia over a 20-year follow-up [11,12]. The U.S. Atherosclerosis Risk in Communities (ARIC) study documented that higher midlife adherence to plant-centered diets rich in fruits, vegetables, legumes, and whole grains was associated with better cognitive outcomes in later life [8]. Similarly, another cohort study analyzing U.K. Biobank data reported that greater adherence to a healthy Japanese dietary pattern, characterized by high consumption of vegetables, seaweed, soy products, and fish, was linked to lower dementia risk [9]. However, a randomized controlled trial in the U.S. of older adults (mean age 70.4 years) demonstrated that cognitive impairment among those randomized to the Mediterranean-DASH Intervention for Neurodegenerative Delay (MIND) diet did not differ significantly from those in the control group [13]. Therefore, it remains unclear whether an association exists in midlife adults. Further, data pertaining to the relation between diet quality and cognitive performance in the context of disadvantaged populations in the U.S. are lacking, particularly among those residing in the Southern U.S. Socioeconomic status, and health behaviors such as smoking are known to influence both dietary choices and cognitive outcomes, particularly in underserved populations. The previously mentioned studies examined general populations at midlife or older ages, with only one specifically measuring the relationship between diet quality and cognition in a rural population [6]. The lack of research is especially concerning when residents of the Southern U.S. have reported the highest prevalences of subjective cognitive decline, particularly in Louisiana (13.6%) [15]. The objective of this study is to examine the relation between overall dietary patterns and cognitive function in midlife adults within the BHS, which is a life-course cohort in a semi-rural Southern U.S. population [16].

## 2. Methods

### 2.1. Participant Methods and Criteria

The Bogalusa Heart Study (BHS) is a prospective cohort study that recruited Black and White children in a semi-rural community located in Louisiana, beginning in 1972, and followed them into adulthood [16]. For this analysis, cross-sectional data from the 2013–2016 survey were examined. The inclusion criteria were the presence of dietary and cognitive data and midlife age, defined as age ≥ 35 years. The maximum age of the study participants included was 58 years old. Participants with missing diet or cognitive variables or under 35 years were excluded. All participants provided written informed consent to participate in the study. Of 1298 participants included in the initial 2013–2016 BHS survey, those with missing data on diet (*n* = 224), cognitive function (*n* = 19), or age below 35 were excluded (*n* = 2). The final sample for this analysis was 1053 participants (Figure 1).

### 2.2. Dietary Quality Scores

Dietary pattern scores were calculated to determine overall diet quality. The Lower Mississippi Nutrition Intervention Research Initiative (Delta NIRI) food frequency questionnaire (FFQ) was used to collect data on participant diet consumption. The Delta NIRI was chosen because it includes foods more commonly found in the Southern regions, including Bogalusa, Louisiana, and has been validated in its use among participants from the lower Mississippi Delta region as part of the Jackson Heart Study [17]. The Delta NIRI has been validated against four 24-h dietary recalls, with macronutrient correlations ranging from 0.33–0.70 in men and 0.30–0.49 in women [17]. In that validation study, the Delta NIRI was on average also able to classify over 30% of individuals into the same dietary intake quartiles as the 24-h dietary recalls with minimal misclassification [17]. From the Delta NIRI, three dietary pattern scores were calculated: the Alternate Healthy Eating Index 2010 (AHEI), the Healthy Eating Index 2015 (HEI), and the Alternate Mediterranean Diet (aMed) scores. Further methodological details on how these scores were derived from the Delta NIRI can be found elsewhere [18].

The Healthy Eating Index is a validated instrument used to measure adherence to the *Dietary Guidelines for Americans*, which are issued every five years by the United States Department of Agriculture (USDA) based on current nutritional science [19]. The HEI consists of 12 components, such as total fruit, vegetables, grains, dairy, protein, seafood, sodium, and empty calories [20]. The total scores of each component are summed to generate a final score [20]. The HEI produces scores ranging from 0 to 100, with 100 indicating that the dietary intake aligns with the guidelines [21]. The AHEI includes similar components as the HEI, except the addition of foods that are shown to be predictors of chronic disease and the removal of some HEI components, such as dairy intake [22]. AHEI scores range from 0 to 110, with 110 indicating perfect adherence to dietary recommendations [22]. In contrast to these dietary indexes, the aMed, which only includes nine components: total fruit and vegetables, nuts, legumes, fish, whole grains, MUFA/SFA ratio, alcohol, and red and processed meats [23]. Each component is designated a score of 1 if the aMed criteria for consumption has been met and 0 otherwise, yielding a scoring range of 0 to 9 [24]. This scoring approach is in contrast to the HEI and AHEI, which are based on pre-specified optimal intake levels.

### 2.3. Cognitive Outcomes

Participants completed a battery of nine validated neuropsychological assessments to evaluate attention, processing speed, episodic memory, executive function, and language abilities. Attention and processing speed were measured using the Digit Span Forward subtest of the Wechsler Adult Intelligence Scale—Fourth Edition (WAIS-IV) and the Trail-Making Test Part A. Episodic memory was assessed with the delayed free recall and recognition subtests of the Wechsler Memory Scale-Fourth Edition (WMS-IV). Executive function was measured by use of the Digit Symbol Coding subtest of the WAIS-IV and the Trail-Making Test Part B. Language abilities were measured with the vocabulary subtest of the WAIS-IV and the word and letter reading subtests of the Wide Range Achievement Test-Fourth Edition (WRAT-IV). Raw scores were standardized into z-scores and averaged to calculate a global cognitive score (GCS), which was used for primary analyses. Further details are available in prior BHS publications [25,26].

### 2.4. Covariates

Participant demographic variables such as age, sex, and race were collected via questionnaires administered. Age was calculated as the subject’s date of taking the BHS survey subtracted by their date of birth. Sex and race were self-reported. Race was not interpreted as an implication of biological differences but rather as an indication of systematic differences and social inequalities. Socioeconomic status (SES) was measured by employment status and education. The highest level of education was categorized as receiving a high school diploma/general educational development (GED) or less education or completing some level of education beyond high school/GED (HS/GED). Participants were categorized as currently active or non-active smokers. Physical activity was assessed using the long-form International Physical Activity Questionnaire (IPAQ), a validated tool used to collect self-reported data on frequency and perceived intensity of physical activity, broken up into categories such as work-related activities, domestic/yard work, leisure activities, and transportation [27]. From the IPAQ, total metabolic equivalent of task (MET)-minutes per week was calculated. Using IPAQ criteria, extreme outliers were treated as missing values [27]. Total energy intake was reported as kilocalories per day (kcal/d). Height (m) and weight (kg) were measured in duplicate and averaged to calculate body mass index (BMI). A BMI greater than or equal to 30 was considered obese [28]. Diabetes was defined as a fasting glucose (mg/dL) greater than or equal to 126 or if participants reported actively taking medication for diabetes [29]. Hypertension was defined as having systolic blood pressure greater than or equal to 130 mmHg, diastolic blood pressure greater than or equal to 80 mmHg, or if participants reported actively taking medication for hypertension [30].

### 2.5. Procedures

Dietary and cognitive assessments were collected as part of the 2013–2016 Bogalusa Heart Study (BHS) visit using standardized instruments. All assessments were administered in-person by trained research staff. Dietary data were collected via a validated 130-item food frequency questionnaire (FFQ), adapted from the Harvard FFQ, which captured typical dietary intake over the past year. Participants completed the FFQ independently or with staff assistance, and data were used to compute diet quality scores (AHEI, HEI, and aMed). Cognitive assessments were administered in a standardized order in a quiet room, typically requiring 60–90 min to complete. Interviewers conducting the cognitive tests were trained according to the established study protocol to ensure consistency and minimize measurement error. Cognitive scores were calculated and reviewed by data managers according to BHS protocols.

### 2.6. Statistical Analysis

Descriptive statistics are presented by quartile of dietary pattern scores. Means and standard deviations were compared with ANOVA to test for significant differences across the quartiles of the dietary pattern scores for continuous variables. Categorical variables, expressed as frequencies, were compared using Pearson’s chi-squared statistics. Skewed variables were log-transformed before analysis. Generalized linear models were constructed to measure the least squares mean estimates of cognitive domain scores and GCS across the levels of dietary pattern adherence. The median of each quartile was treated as a continuous variable to analyze trends. For the outcome of overall cognitive profile, multinomial logistic regression was employed to calculate odds ratios (ORs) of cognition profiles with 95% confidence intervals (CI). Although odds ratios were used in this cross-sectional analysis, we acknowledge that prevalence ratios may provide more interpretable effect estimates. ORs were selected due to their compatibility with multinomial and logistic models and are interpreted with caution. Cognitive profile was also treated as a dichotomous outcome to test the likelihood of low cognition compared to those who were not categorized with low cognition. Likelihood per standard deviation (SD) increase was also calculated by treating the standardized dietary score as a continuous variable. Complete case analysis was conducted under the assumption that missing covariate data was missing completely at random to ensure unbiased estimates.

Model 1 was unadjusted. Model 2 was adjusted for demographic characteristics including age, sex (male/female), and race/ethnicity (White/Black) [5,11,12]. Model 3 was additionally adjusted for employment (employed/not employed), education (≤high school diploma/GED, >high school diploma/GED), and health behaviors including smoking (active/not active), total energy intake (kcal/d), and MET-minutes per week [4,7,8]. Model 4 was further adjusted for BMI (continuous), diabetes (yes/no), and hypertension (yes/no) to consider potential confounding by comorbidities [4,5,7,8,11]. Interaction multiplicative product terms were added to Model 4 to assess the presence of effect modifiers and explore potential mechanisms involved in the relationship between dietary adherence and cognition by multiplying the covariate by the continuous dietary pattern score. When statistically significant interaction was present, a joint analysis was performed to examine the combined effect of the modifier and the dietary quartiles on the dependent variable. A *p*-value less than 0.05 was considered statistically significant. SAS, version 9.4 (SAS Institute Inc., Cary, NC, USA), and R, version 4.3.1 (R Foundation for Statistical Computing, Vienna, Austria), were used.

## 3. Results

### 3.1. Participant Characteristics

The mean age of the 1053 participants included in the analysis was 48.18 (SD = 5.22) years. The minimum age was 35 years, and the maximum age was 58 years. In the sample, 38.94% of the sample identified as male, and 31.14% identified as Black/African-American. Table 1 shows participant characteristics according to dietary pattern quartiles. The median BMI among participants was 30.92, with a range from 16.36 to 67.74, indicating that while obesity was common in the sample, it was not an inclusion criterion.

### 3.2. Association Between Diet Quality and Global Cognitive Score

The least squares mean estimates with 95% CIs for the three cognitive performance domains (attention and processing, episodic memory, and executive function) as well as GCS are presented in Table 2 for each dietary pattern. In an unadjusted analysis, greater adherence to AHEI and HEI was associated with greater GCS (*P*-trend = 0.03 for AHEI, 0.02 for HEI), but no significant association was observed between aMed and GCS (*P*-trend = 0.08). When adjusting for Model 4 covariates, the associations of higher AHEI, HEI, and aMed with higher GCSs were observed (*P*-trend = 0.01 for AHEI, 0.02 for HEI, and 0.04 for aMed). Higher adherence to AHEI, HEI, or aMed was associated with higher episodic memory scores after adjustment for Model 4 covariates (*P*-trend = 0.02 for AHEI, 0.03 for HEI, 0.02 for aMed). In the unadjusted analysis of executive function, higher adherence to AHEI and HEI yielded the highest mean estimate (*P*-trend = 0.01 for AHEI, 0.004 for HEI), and significant associations across all dietary patterns were only observed when adjusting for age, sex, and race (*P*-trend = 0.001 for AHEI, 0.003 for HEI, 0.04 for aMed).

### 3.3. Comparing Low, Average, and Optimal Neuropsychological Profiles According to Diet Quality

Based on the GCS, 219 of the 1053 participants were categorized as having low overall cognitive performance, 513 average, and 321 optimal. Table 3 presents the multinomial odds ratios with 95% CIs of low and average cognition per dietary quartile compared to the lowest adherence quartile. Figure 2 is the forest plot of the Model 4 results. For AHEI, aMed, and HEI, no significant association between adherence and cognition performance was observed in unadjusted analyses (Table 3). With adjustment for age, sex, race, employment, education, smoking, total energy intake, MET-minutes, and comorbidities, higher diet quality was associated with lower odds of having lower cognitive performance (*P*-trend = 0.02 for AHEI, 0.02 for HEI, and 0.01 for aMed) (Table 3). No significant associations between dietary patterns and average cognition were observed when adjusting for similar covariates (Table 3).

### 3.4. Comparing Low and Other Neuropsychological Profiles According to Diet Quality

Table 4 and Figure 3 show the dichotomous outcome (low and other neuropsychological profiles) odds ratios. In unadjusted analyses, no significant associations between diet quality (AHEI, aMed, or HEI) and cognitive performance were observed (Table 4), but participants with higher adherence to AHEI and aMed had the lowest odds of lower cognitive performance when adjusting for age, sex, race, employment, education, smoking status, total energy intake, MET-minutes, BMI, diabetes, and hypertension (*P*-trend = 0.03 for AHEI, 0.03 for aMed).

### 3.5. Interaction Testing

The associations between dietary adherence and GCS were not modified by age, sex, race, education, employment, smoking status, total energy intake, physical activity, BMI, diabetes, or hypertension (all *P*-interaction > 0.05; Appendix A). The associations between dietary scores and likelihood of low cognitive performance in multinomial regression were not modified by age, sex, race, education, smoking status, total energy intake, physical activity, BMI, diabetes, or hypertension (all *P*-interaction > 0.05; Appendix A). However, the association between dietary adherence and odds of low cognitive performance was found to be modified by employment status (*P*-interaction = 0.28 for AHEI, 0.01 for HEI, and 0.11 for aMed).

### 3.6. Joint Analysis

Due to potential modification of employment on the association between diet and cognitive profile, joint analysis was conducted to further investigate the effect of employment status (Appendix A). Participants were categorized into groups based on their dietary quartile and employment status, with employed persons in the highest quartile of the respective dietary pattern as the reference group. Joint analysis revealed that participants that were unemployed and in the lowest quartile of AHEI and aMed were highly associated with low cognitive performance (*P*-trend = 0.02 for AHEI and 0.01 for aMed). With HEI, participants in Q2 observed the greatest association with low cognitive performance if they were unemployed (*P*-trend = 0.02).

## 4. Discussion

Greater diet quality was associated with better cognitive performance among midlife adults. Participants in the highest quartiles of AHEI, HEI, and aMed had the highest global cognitive scores when adjusting for age, sex, race, employment, education, smoking status, total energy intake, physical activity, BMI, diabetes, and hypertension. Consequently, when comparing low, average, and optimal neurological profiles, participants with the highest adherence to AHEI and aMed, and those in the third highest quartile of HEI, had the lowest association with lower cognitive performance when adjusting for the same covariates. When comparing odds of low cognitive profile to other profiles, statistically significant trends were present for AHEI and aMed, suggesting that higher adherence to these dietary patterns had the lowest association with a low cognitive performance when adjusting for participant demographics, employment, education, and comorbidity covariates. However, no significant trends were detected in the association between HEI and lower cognitive performance in dichotomous regression. The combined effect of employment and dietary quartile on the observed association suggested that those who were unemployed and in the lowest quartiles of AHEI and aMed were associated with lower cognitive performance.

The overall findings are consistent with previous observational studies that have demonstrated adherence to a healthy diet is associated with higher cognitive function [8,9,10,11,12,13,31,32,33]. Several large cohort studies have focused specifically on the Mediterranean diet, DASH, and the Alternate Healthy Eating Index (AHEI). In the U.S. Nurses’ Health Study, long-term adherence to these diets—each emphasizing fruits, vegetables, whole grains, nuts, and healthy fats—was associated with a lower risk of subjective cognitive decline [10]. The prospective U.K. Whitehall II study and the Swedish Malmö Diet and Cancer Study both reported that better midlife diet quality, particularly patterns aligned with AHEI components, was linked to a lower risk of developing dementia over a 20-year follow-up [11,12]. The U.S. Atherosclerosis Risk in Communities (ARIC) study documented that higher midlife adherence to plant-centered diets rich in fruits, vegetables, legumes, and whole grains was associated with better cognitive outcomes in later life [8]. Similarly, another cohort study analyzing U.K. Biobank data reported that greater adherence to a healthy Japanese dietary pattern, characterized by high consumption of vegetables, seaweed, soy products, and fish, was linked to lower dementia risk [9]. However, not all studies have found consistent associations. A 2023 randomized controlled trial in the U.S. of older adults (mean age 70.4 years) demonstrated that cognitive impairment among those randomized to the Mediterranean-DASH Intervention for Neurodegenerative Delay (MIND) diet did not differ significantly from those in the control group [13]. Another 2022 randomized controlled trial reported no significant cognitive benefits from omega-3 supplementation in older adults over a 2-year period, suggesting that the efficacy of such interventions may depend on factors like baseline nutritional status, dosage, and duration [33].

Recent studies further support the potential role of specific dietary components in cognitive health. For instance, a cross-sectional analysis from the Framingham Heart Study reported that midlife adults with higher red blood cell omega-3 fatty acid levels had larger hippocampal volumes and better cognitive performance, including abstract reasoning, compared to those with lower levels [34]. Similarly, a prospective cohort study found that higher whole grain intake was associated with slower cognitive decline over a 6-year period, especially in episodic memory and perceptual speed, with the strongest associations observed among African American participants [35].

Existing research has studied the potential underlying mechanisms by which diet can impact cognition. Nutrigenomic studies have examined the relationship between the gut microbiota found in the gastrointestinal (GI) tract and its impact on cognition via the gut-brain axis. The gut microbiota is responsible for the production of short-chain fatty acids that are responsible for regulating gene expression, as well as the production of inflammatory cytokines [36]. Disruption to the microbiota, also known as dysbiosis, can increase the risk of chronic diseases and dysfunction of the microbiota’s metabolic properties. The GI tract is responsible for producing hormones and signaling molecules, such as those that regulate appetite, mood, memory, and stress [37]. The GI tract can also synthesize neurotransmitters responsible for the stimulation of brain functions, such as dopamine, epinephrine, and acetylcholine [37]. A balanced diet can influence the stability of the gut microbiota and, in turn, the effect it has on cognition. Carbohydrates, for example, can act as either a protective or antagonistic factor depending on the subtype [38]. Simple carbohydrates, sugars, have been found to be associated with worse global cognition, whereas fiber-rich diets have been shown to benefit memory and overall cognitive performance [38]. Consumption of dietary fats such as saturated fatty acids and monounsaturated fats has been linked to poor performance in learning and memory, unlike polyunsaturated fats and omega-3 intake, which have been shown to benefit memory and processing speed [38]. Protein consumption has yielded mixed results in older populations, as there is evidence that it may be beneficial for memory and global cognitive functioning, but overconsumption and lower-quality protein can be detrimental [14,34,35]. Adhering to a healthy diet, such as those suggested by the USDA, not only promotes a healthy gut microbiota but also prevents the development of adverse cardiometabolic risk profiles, such as high BMI, hypertension, and diabetes, which also protects against cognitive decline and can reduce the risk for chronic illnesses, including neurodegenerative diseases [39,40]. Additionally, deficiencies in key micronutrients, such as vitamin B12, vitamin D, and iron, may further exacerbate cognitive impairment by disrupting essential neurological and metabolic processes [41].

There are several strengths to this study. Although this study does not compare the dietary patterns to each other, the findings suggest that high adherence to dietary guidelines is associated with reduced cognitive decline. Another strength of this study is the employment of a validated, culturally specific FFQ to determine diet quality and dietary guideline adherence. The previous validation of the Delta NIRI among individuals from the same region as BHS participants allowed for a very detailed and reliable measurement of dietary intake provided by the participants. The cognitive measurements were also collected using validated tools, which allowed for analysis of a global cognitive score, as well as subdomains within cognitive function to examine which components of the global cognitive score were most impacted by diet. This study is not without its limitations, however. Due to the cross-sectional design, causal relationships cannot be established, and temporality between diet and cognition remains uncertain. We cannot conclude that a poor-quality diet over an extended period of time is a direct cause of reduced cognitive function in midlife adults, but the results indicate there may be some linkage present. Another limitation is sample size. The observed associations between dietary quartiles and lower cognition may be attributable to inadequate sample sizes within the quartiles and clusters of cognition. This may have also impacted the stratified analysis of variables such as employment, where unemployment was found to be associated with a lower likelihood of low cognition compared to those who were employed. Results should be interpreted with caution, as the power for this study was likely limited by sample size. Thirdly, several of the examined factors were obtained via self-report. The reliability and accuracy of some measurements, such as MET-minutes per week, may have been over- or under-estimated, as some values were implausible. A key limitation of this study is the relatively short follow-up period, particularly in the context of Alzheimer's disease development. Future studies with extended monitoring periods are needed to validate our findings and assess the persistence of these associations over time. We also recommend that future studies further examine the impact of dietary guideline adherence and cognitive function in midlife adults living in rural communities. Longitudinal studies with underrepresented populations would allow for exploration of casual associations and provide evidence pertaining to the potential linkage between dietary guideline adherence and cognition, as well as contributing factors.

## 5. Conclusions

In conclusion, greater diet quality was associated with better cognitive performance among midlife adults in the BHS. These findings underscore the need for further research on dietary interventions during midlife. Future studies should employ longitudinal designs and include underrepresented and socioeconomically diverse populations to clarify the temporal relationships between diet and cognitive health. These results suggest that promoting high-quality diets among midlife adults could serve as an effective public health intervention to improve cognitive outcomes and reduce long-term disease burden.

## Figures and Tables

**Figure 1 nutrients-17-01636-f001:**
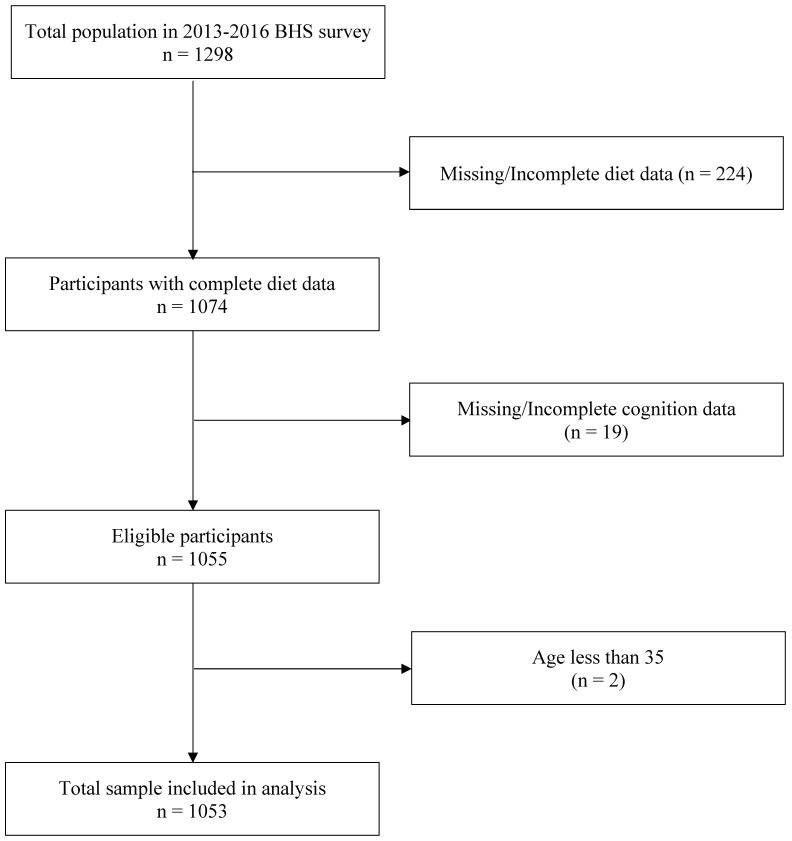
Participant criteria flowchart.

**Figure 2 nutrients-17-01636-f002:**
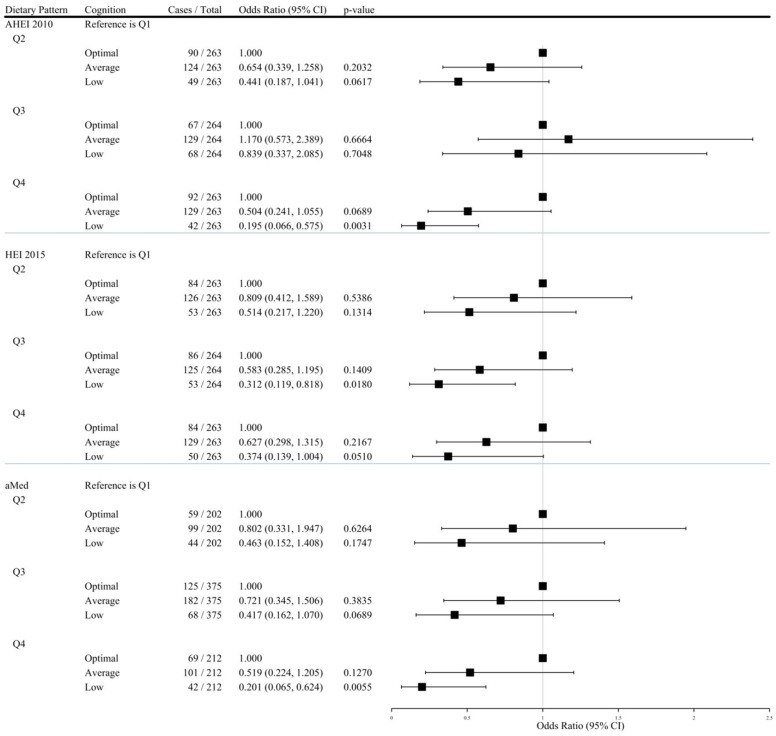
Adjusted Odds Ratios (95% CI) of Having Low or Average Cognition Across Dietary Quartiles (N = 1053).

**Figure 3 nutrients-17-01636-f003:**
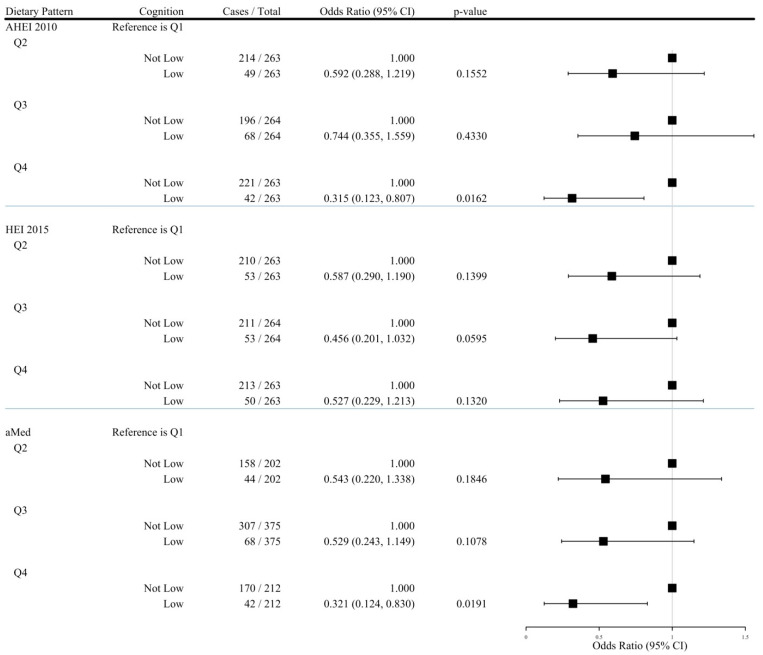
Adjusted Odds Ratios (95% CI) of Low Cognition Across Dietary Quartiles (N = 1053).

**Table 1 nutrients-17-01636-t001:** Participant Characteristics According to Dietary Pattern Quartiles in the Bogalusa Heart Study 2013–2016 (N = 1053).

Characteristic	Q1	Q2	Q3	Q4	*p*-Value ^1^
**Alternate Healthy Eating Index 2010**					
	*n* = 263	*n* = 263	*n* = 264	*n* = 263	
Median Score (95% CI)	33.82 (25.49–37.49)	41.39 (38.36–44.04)	48.59 (44.61–51.51)	56.65 (52.27–69.55)	
**Age (years)** ^2^	47.25 ± 5.26	47.62 ± 5.29	48.60 ± 5.18	49.24 ± 4.94	<0.0001
**Sex** ^3^					0.4181
Male	107 (40.68)	107 (40.68)	105 (39.77)	91 (34.60)	
Female	156 (59.32)	156 (59.32)	159 (60.23)	172 (65.40)	
**Race** ^3^					0.0174
White	200 (76.05)	182 (69.47)	170 (64.64)	171 (65.27)	
Black	63 (23.95)	80 (30.53)	93 (35.36)	91 (34.73)	
**Highest Education** ^3^					0.0008
≤High School/GED ^4^	199 (75.95)	178 (67.94)	191 (72.35)	159 (60.46)	
>High School/GED	63 (24.05)	84 (32.06)	73 (27.65)	104 (39.54)	
**Employed** ^3^	171 (65.52)	166 (63.36)	173 (65.53)	185 (70.61)	0.3433
**MET-minutes/week** ^2,5^	3787.44 ± 6310.50	3418.79 ± 5719.28	3346.42 ± 5702.61	2881.33 ± 5359.35	0.9194
**Total Energy Intake (kcal/d)** ^2^	2846.58 ± 531.35	2809.04 ± 539.79	2818.50 ± 566.17	2697.13 ± 479.54	0.1515
**BMI (kg/m^2^)** ^2,6^	30.93 ± 7.24	31.75 ± 7.90	31.98 ± 7.65	30.95 ± 7.77	0.2618
**Diabetes** ^3^	42 (15.97)	40 (15.21)	50 (18.94)	41 (15.59)	0.6431
**Hypertension** ^3^	129 (49.05)	127 (48.29)	108 (40.91)	117 (44.49)	0.2125
**Smoking Status** ^3^					<0.0001
Active Smoker	67 (25.48)	53 (20.15)	35 (13.26)	22 (8.37)	
Not Active Smoker	196 (74.52)	210 (79.85)	229 (86.74)	241 (91.63)	
**Healthy Eating Index 2015**					
	*n* = 263	*n* = 263	*n* = 264	*n* = 263	
Median Score (95% CI)	48.57 (40.09–52.66)	56.17 (53.38–58.77)	62.22 (59.25–64.98)	70.11 (65.95–80.09)	
**Age (years)** ^2^	47.26 ± 5.36	47.45 ± 5.26	48.85 ± 5.22	49.15 ± 4.80	<0.0001
**Sex** ^3^					0.0010
Male	126 (47.91)	100 (38.02)	103 (39.02)	81 (30.80)	
Female	137 (52.09)	163 (61.98)	161 (60.98)	182 (69.20)	
**Race** ^3^					0.0940
White	191 (72.62)	190 (72.24)	170 (64.64)	172 (65.90)	
Black	72 (27.38)	73 (27.76)	93 (35.36)	89 (34.10)	
**Highest Education** ^3^					0.0007
≤High School/GED ^4^	202 (76.81)	186 (70.72)	182 (68.94)	159 (60.46)	
>High School/GED	61 (23.37)	77 (29.28)	82 (31.06)	104 (39.54)	
**Employed** ^3^	169 (64.26)	173 (66.28)	174 (66.16)	179 (68.32)	0.8085
**MET-minutes/week** ^2,5^	3653.19 ± 6265.86	2885.96 ± 5339.81	2298.98 ± 5589.17	3495.64 ± 5903.53	0.3043
**Total Energy Intake (kcal/d)** ^2^	2851.60 ± 531.35	2856.43 ± 561.13	2751.06 ± 519.01	2722.00 ± 505.84	0.0974
**BMI (kg/m^2^)** ^2,6^	30.95 ± 7.59	31.77 ± 7.82	31.95 ± 7.67	30.94 ± 7.47	0.2733
**Diabetes** ^3^	42 (15.97)	40 (15.21)	55 (16.67)	47 (17.87)	0.8651
**Hypertension** ^3^	130 (49.43)	114 (43.35)	129 (48.86)	108 (41.06)	0.1445
**Smoking Status** ^3^					<0.0001
Active Smoker	67 (25.48)	51 (19.39)	38 (14.39)	21 (7.98)	
Not Active Smoker	196 (74.52)	212 (80.61)	226 (85.61)	242 (92.02)	
**Alternate Mediterranean Diet**					
	*n* = 264	*n* = 202	*n* = 375	*n* = 212	
Median Score (95% CI)	2.00 (1.00–2.00)	3.00 (3.00–3.00)	4.00 (4.00–5.00)	6.00 (6.00–8.00)	
**Age (years)** ^2^	47.48 ± 5.25	48.08 ± 5.27	48.21 ± 5.22	49.09 ± 5.04	0.0101
**Sex** ^3^					0.3184
Male	114 (43.18)	77 (38.12)	145 (38.67)	74 (34.91)	
Female	150 (56.82)	125 (61.88)	230 (61.33)	138 (65.09)	
**Race** ^3^					0.0064
White	201 (76.14)	142 (70.30)	249 (66.94)	131 (61.79)	
Black	63 (23.86)	60 (29.70)	123 (33.06)	81 (38.21)	
**Highest Education** ^3^					0.0831
≤High School/GED ^4^	195 (74.43)	143 (70.79)	252 (67.20)	137 (64.62)	
>High School/GED	67 (25.57)	59 (29.21)	123 (32.80)	75 (35.38)	
**Employed** ^3^	162 (61.83)	133 (66.17)	262 (70.05)	138 (65.09)	0.1843
**MET-minutes/week** ^2,5^	3147.74 ± 5855.73	3738.57 ± 6087.07	3340.59 ± 5853.87	3290.42 ± 5273.09	0.1346
**Total Energy Intake (kcal/d)** ^2^	2703.92 ± 479.01	2768.88 ± 541.24	2784.71 ± 537.17	2909.24 ± 542.61	0.0293
**BMI (kg/m^2^)** ^2,6^	30.48 ± 7.13	31.90 ± 7.66	31.91 ± 8.19	31.19 ± 7.18	0.0893
**Diabetes** ^3^	37 (14.02)	35 (17.33)	64 (17.07)	37 (17.45)	0.6795
**Hypertension** ^3^	107 (40.53)	106 (52.48)	175 (46.67)	93 (43.87)	0.0716
**Smoking Status** ^3^					0.0006
Active Smoker	66 (25.00)	31 (15.35)	60 (16.00)	20 (9.43)	
Not Active Smoker	198 (75.00)	171 (84.65)	315 (84.00)	192 (90.57)	

^1^ *p*-value from comparison across quartiles: ANOVA for continuous variables, Pearson chi-squared for categorical variables; ^2^ mean ± standard deviation; ^3^ N (%); ^4^ GED, general educational development; ^5^ MET, total metabolic equivalent of task; ^6^ BMI, body mass index.

**Table 2 nutrients-17-01636-t002:** Least Squares Mean Estimates (95% CI) of Cognitive Performance Outcomes by Quartile of Dietary Score (N = 1053).

Cognitive Outcome	Q1 ^1^	Q2 ^1^	Q3 ^1^	Q4 ^1^	*P* _trend_ ^2^
**Alternate Healthy Eating Index 2010**					
	*n* = 263	*n* = 263	*n* = 264	*n* = 263	
**Global Cognitive Score**					
Model 1	0.214 (−0.358, 0.786)	0.865 (0.293, 1.437) ***	−0.132 (−0.703, 0.439)	1.491 (0.919, 2.063) ***	0.0258
Model 2	−0.943 (−1.496, −0.389) ***	−0.024 (−0.567, 0.519)	−0.809 (−1.346, −0.273) ***	0.800 (0.256, 1.343) ***	0.0002
Model 3	−0.270 (−1.107, 0.566)	0.715 (−0.156, 1.587)	−0.116 (−1.077, 0.845)	1.589 (0.580, 2.598) ***	0.0182
Model 4	−0.174 (−1.081, 0.733)	0.830 (−0.128, 1.788)	−0.016 (−1.051, 1.020)	1.758 (0.069, 2.847) ***	0.0144
**Attention and Processing**					
Model 1	0.054 (−0.128, 0.236)	0.172 (−0.010, 0.354)	−0.113 (−0.295, 0.068)	0.275 (0.093, 0.457) ***	0.3363
Model 2	−0.022 (−0.213, 0.168)	0.108 (−0.079, 0.295)	−0.156 (−0.341, 0.028)	0.243 (0.056, 0.430) *	0.1838
Model 3	−0.0003 (−0.284, 0.284)	0.316 (0.020, 0.612) *	0.008 (−0.318, 0.334)	0.390 (0.048, 0.733) *	0.1943
Model 4	−0.050 (−0.357, 0.257)	0.255 (−0.070, 0.579)	−0.022 (−0.373, 0.329)	0.339 (−0.030, 0.708)	0.1785
**Episodic Memory**					
Model 1	−0.053 (−0.260, 0.155)	0.215 (0.008, 0.423) *	0.073 (−0.134, 0.280)	0.411 (0.203, 0.618) ***	0.0078
Model 2	−0.381 (−0.590, −0.173) ***	−0.032 (−0.236, 0.173)	−0.104 (−0.306, 0.099)	0.235 (0.030, 0.440) *	0.0001
Model 3	−0.245 (−0.600, 0.111)	−0.011 (−0.381, 0.359)	0.031 (−0.378, 0.439)	0.316 (−0.113, 0.744)	0.0349
Model 4	−0.176 (−0.561, 0.209)	0.067 (−0.339, 0.474)	0.118 (−0.321, 0.557)	0.433 (−0.029, 0.895)	0.0224
**Executive Function**					
Model 1	−0.044 (−0.234, 0.145)	0.177 (−0.013, 0.366)	−0.009 (−0.198, 0.180)	0.369 (0.180, 0.559) ***	0.0131
Model 2	−0.277 (−0.469, −0.084) ***	−0.014 (−0.204, 0.175)	−0.147 (−0.334, 0.040)	0.220 (0.030, 0.409) *	0.0013
Model 3	−0.164 (−0.450, 0.123)	0.254 (−0.045, 0.552)	−0.047 (−0.376, 0.282)	0.185 (−0.160, 0.531)	0.2253
Model 4	−0.149 (−0.458, 0.160)	0.263 (−0.064, 0.590)	0.004 (−0.350, 0.357)	0.210 (−0.162, 0.581)	0.1802
**Healthy Eating Index 2015**					
	*n* = 263	*n* = 263	*n* = 264	*n* = 263	
**Global Cognitive Score**					
Model 1	0.017 (−0.559, 0.592)	0.669 (0.094, 1.245) *	0.761 (0.187, 1.336) **	0.987 (0.412, 1.563) ***	0.0215
Model 2	−0.937 (−1.487, −0.388) ***	−0.344 (−0.896, 0.208)	0.096 (−0.445, 0.637)	0.233 (−0.320, 0.786)	0.0016
Model 3	−0.143 (−1.008, 0.721)	0.297 (−0.579, 1.173)	0.607 (−0.356, 1.569)	1.195 (0.201, 2.189) *	0.0256
Model 4	−0.081 (−1.030, 0.868)	0.337 (−0.645, 1.320)	0.654 (−0.376, 1.684)	1.289 (0.238, 2.340) *	0.0238
**Attention and Processing**					
Model 1	−0.025 (−0.208, 0.157)	0.080 (−0.103, 0.262)	0.181 (−0.001, 0.363)	0.151 (−0.031, 0.334)	0.1360
Model 2	−0.099 (−0.287, 0.090)	0.014 (−0.175, 0.203)	0.140 (−0.046, 0.325)	0.111 (−0.078, 0.301)	0.0801
Model 3	0.022 (−0.269, 0.313)	0.033 (−0.262, 0.328)	0.414 (0.090, 0.738) *	0.303 (−0.032, 0.638)	0.0721
Model 4	−0.063 (−0.381, 0.255)	−0.032 (−0.362, 0.298)	0.353 (0.008, 0.698) *	0.252 (−0.102, 0.604)	0.0502
**Episodic Memory**					
Model 1	−0.040 (−0.248, 0.168)	0.177 (−0.031, 0.385)	0.249 (0.041, 0.457) *	0.260 (0.052, 0.468) *	0.0409
Model 2	−0.304 (−0.511, −0.098) ***	−0.110 (−0.317, 0.098)	0.081 (−0.122, 0.285)	0.069 (−0.140, 0.276)	0.0056
Model 3	−0.260 (−0.623, 0.104)	−0.103 (−0.472, 0.266)	0.227 (−0.178, 0.632)	0.229 (−0.189, 0.647)	0.0294
Model 4	−0.201 (−0.599, 0.198)	−0.042 (−0.455, 0.371)	0.284 (−0.149, 0.716)	0.312 (−0.130, 0.754)	0.0247
**Executive Function**					
Model 1	−0.042 (−0.232, 0.148)	0.060 (−0.129, 0.250)	0.121 (−0.068, 0.310)	0.352 (0.163, 0.542) ***	0.0038
Model 2	−0.214 (−0.405, 0.023) *	−0.154 (−0.346, 0.037)	−0.012 (−0.199, 0.176)	0.173 (−0.019, 0.365)	0.0025
Model 3	−0.084 (−0.379, 0.211)	−0.011 (−0.310, 0.288)	0.197 (−0.132, 0.526)	0.152 (−0.188, 0.491)	0.1724
Model 4	−0.099 (−0.421, 0.223)	0.016 (−0.318, 0.349)	0.210 (−0.139, 0.560)	0.167 (−0.190, 0.523)	0.1342
**Alternate Mediterranean Diet**					
	*n* = 264	*n* = 202	*n* = 375	*n* = 212	
**Global Cognitive Score**					
Model 1	−0.072 (−0.646, 0.502)	0.844 (0.188, 1.500) *	0.832 (0.402, 1.365) ***	0.746 (0.106, 1.387) *	0.0842
Model 2	−1.210 (−1.763, −0.658) ***	−0.083 (−0.701, 0.536)	0.091 (−0.369, 0.551)	0.170 (−0.431, 0.770)	0.0012
Model 3	−0.360 (−1.414, 0.694)	0.415 (−0.672, 1.502)	0.416 (−0.341, 1.174)	1.087 (0.126, 2.048) *	0.0436
Model 4	−0.319 (−1.477, 0.840)	0.493 (−0.691, 1.677)	0.430 (−0.408, 1.268)	1.156 (0.141, 2.171) *	0.0428
**Attention and Processing**					
Model 1	−0.024 (−0.206, 0.158)	0.152 (−0.056, 0.361)	0.124 (−0.029, 0.277)	0.146 (−0.057, 0.349)	0.2807
Model 2	−0.105 (−0.295, 0.085)	0.093 (−0.119, 0.306)	0.070 (−0.088, 0.228)	0.118 (−0.089, 0.324)	0.1502
Model 3	−0.137 (−0.492, 0.218)	0.185 (−0.182, 0.551)	0.210 (−0.045, 0.466)	0.351 (0.027, 0.675) *	0.0545
Model 4	−0.220 (−0.609, 0.168)	0.062 (−0.335, 0.460)	0.156 (−0.126, 0.437)	0.276 (−0.065, 0.617)	0.0432
**Episodic Memory**					
Model 1	−0.059 (−0.266, 0.149)	0.112 (−0.125, 0.349)	0.286 (0.112, 0.460) ***	0.262 (0.031, 0.494) *	0.0290
Model 2	−0.372 (−0.580, −0.164) ***	−0.141 (−0.374, 0.092)	0.073 (−0.100, 0.246)	0.116 (−0.109, 0.342)	0.0008
Model 3	−0.422 (−0.864, 0.021)	−0.088 (−0.545, 0.369)	0.047 (−0.272, 0.365)	0.301 (−0.103, 0.704)	0.0135
Model 4	−0.361 (−0.845, 0.125)	−0.033 (−0.530, 0.464)	0.084 (−0.268, 0.436)	0.358 (−0.068, 0.784)	0.0145
**Executive Function**					
Model 1	−0.029 (−0.219, 0.161)	0.111 (−0.106, 0.328)	0.208 (0.049, 0.367) *	0.173 (−0.038, 0.385)	0.1415
Model 2	−0.241 (−0.434, −0.049) *	−0.078 (−0.294, 0.138)	0.033 (−0.127, 0.193)	0.043 (−0.166, 0.252)	0.0380
Model 3	−0.174 (−0.533, 0.186)	0.021 (−0.350, 0.392)	0.064 (−0.194, 0.323)	0.223 (−0.105, 0.551)	0.0981
Model 4	−0.161 (−0.553, 0.232)	−0.030 (−0.432, 0.372)	0.084 (−0.201, 0.368)	0.206 (−0.139, 0.550)	0.1086

^1^ Least squares mean estimate (95% CI); ^2^ *p*-value for linear trend calculated using the median from each quartile and treating it as a continuous variable; * *p* < 0.05, ** *p* < 0.01, *** *p* < 0.005; Model 1: Unadjusted. Model 2: Model 1, age, sex, race. Model 3: Model 2, employment, education (≤high school/GED, >high school/GED), smoking (active smoker, not active smoker), total energy intake, MET-minutes/week. Model 4: Model 3, BMI, diabetes, hypertension.

**Table 3 nutrients-17-01636-t003:** Adjusted Odds Ratios (95% CI) of Having Low or Average Cognition Across Dietary Quartiles (N = 1053).

	Q1 ^1^	Q2 ^2^	Q3 ^2^	Q4 ^2^	P_Trend_ ^3^	Per SD Increase ^4^
**Alternate Healthy Eating Index 2010**						
**Optimal Cognition** ^1^						
# Cases/Total	72/263	90/263	67/264	92/263		SD = 9.966
	1.000	1.000	1.000	1.000	–	–
**Average Cognition vs. Optimal Cognition**						
# Cases/Total	131/263	124/263	129/264	129/263		SD = 9.966
Model 1	1.000	0.757 (0.510, 1.125)	1.058 (0.701, 1.598)	0.771 (0.520, 1.141)	0.4415	0.850 (0.715, 1.012)
Model 2	1.000	0.701 (0.467, 1.052)	0.926 (0.607, 1.415)	0.637 (0.424, 0.958) *	0.0916	0.901 (0.779, 1.042)
Model 3	1.000	0.668 (0.349, 1.278)	1.225 (0.602, 2.490)	0.537 (0.260, 1.109)	0.2933	0.926 (0.724, 1.185)
Model 4	1.000	0.654 (0.339, 1.258)	1.170 (0.573, 2.389)	0.504 (0.241, 1.055)	0.2247	0.910 (0.708, 1.170)
**Low Cognition vs. Optimal Cognition**						
# Cases/Total	60/263	49/263	68/264	42/263		SD = 9.966
Model 1	1.000	0.653 (0.401, 1.065)	1.218 (0.753, 1.970)	0.548 (0.332, 0.904) *	0.1419	0.969 (0.843, 1.114)
Model 2	1.000	0.535 (0.318, 0.900) *	0.940 (0.563, 1.568)	0.382 (0.222, 0.657) ***	0.0072	0.739 (0.611, 0.894) ^†††^
Model 3	1.000	0.452 (0.193, 1.057)	0.893 (0.361, 2.206)	0.216 (0.074, 0.631) ***	0.0247	0.691 (0.490, 0.974) ^†^
Model 4	1.000	0.441 (0.187, 1.041)	0.839 (0.337, 2.085)	0.195 (0.066, 0.575) ***	0.0152	0.667 (0.471, 0.945) ^†^
**Healthy Eating Index 2015**						
**Optimal Cognition** ^1^						
# Cases/Total	67/263	84/263	86/264	84/263		SD = 9.192
	1.000	1.000	1.000	1.000	–	–
**Average Cognition vs. Optimal Cognition**						
# Cases/Total	133/263	126/263	125/264	129/263		SD = 9.192
Model 1	1.000	0.756 (0.505, 1.131)	0.732 (0.490, 1.095)	0.774 (0.517, 1.157)	0.2317	0.928 (0.807, 1.067)
Model 2	1.000	0.756 (0.500, 1.142)	0.647 (0.426, 0.982) *	0.703 (0.461, 1.069)	0.0825	0.875 (0.756, 1.013)
Model 3	1.000	0.838 (0.429, 1.634)	0.595 (0.291, 1.215)	0.648 (0.311, 1.349)	0.1780	0.763 (0.586, 0.994) ^†^
Model 4	1.000	0.809 (0.412, 1.589)	0.583 (0.285, 1.195)	0.627 (0.298, 1.315)	0.1583	0.752 (0.575, 0.984) ^†^
**Low Cognition vs. Optimal Cognition**						
# Cases/Total	63/263	53/263	53/264	50/263		SD = 9.192
Model 1	1.000	0.671 (0.413, 1.091)	0.655 (0.403, 1.065)	0.633 (0.388, 1.034)	0.0783	0.887 (0.746, 1.053)
Model 2	1.000	0.672 (0.402, 1.123)	0.521 (0.309, 0.877) *	0.541 (0.318, 0.920) *	0.0159	0.797 (0.660, 0.962) ^†^
Model 3	1.000	0.537 (0.229, 1.263)	0.322 (0.123, 0.840) *	0.408 (0.154, 1.081)	0.0336	0.649 (0.457, 0.921) ^†^
Model 4	1.000	0.514 (0.217, 1.220)	0.312 (0.119, 0.818) *	0.374 (0.139, 1.004)	0.0241	0.627 (0.440, 0.895) ^†^
**Alternate Mediterranean Diet**						
**Optimal Cognition** ^1^						
# Cases/Total	68/264	59/202	125/375	69/212		SD = 1.778
	1.000	1.000	1.000	1.000	–	–
**Average Cognition vs. Optimal Cognition**						
# Cases/Total	131/264	99/202	182/375	101/212		SD = 1.778
Model 1	1.000	0.871 (0.563, 1.346)	0.756 (0.522, 1.095)	0.760 (0.497, 1.161)	0.1811	0.931 (0.810, 1.070)
Model 2	1.000	0.796 (0.510, 1.243)	0.657 (0.448, 0.964) *	0.609 (0.392, 0.947) *	0.0213	0.864 (0.748, 0.999) ^†^
Model 3	1.000	0.783 (0.324, 1.891)	0.701 (0.337, 1.458)	0.506 (0.222, 1.162)	0.0965	0.820 (0.628, 1.070)
Model 4	1.000	0.802 (0.331, 1.947)	0.721 (0.345, 1.506)	0.519 (0.224, 1.205)	0.1116	0.823 (0.629, 1.078)
**Low Cognition vs. Optimal Cognition**						
# Cases/Total	65/264	44/202	68/375	42/212		SD = 1.778
Model 1	1.000	0.780 (0.465, 1.309)	0.569 (0.363, 0.893) *	0.637 (0.381, 1.063)	0.0554	0.843 (0.709, 1.003)
Model 2	1.000	0.669 (0.386, 1.159)	0.439 (0.271, 0.711) ***	0.428 (0.245, 0.746) ***	0.0014	0.734 (0.608, 0.886) ^†††^
Model 3	1.000	0.436 (0.145, 1.310)	0.414 (0.163, 1.053)	0.203 (0.066, 0.621) **	0.0079	0.587 (0.407, 0.847) ^†††^
Model 4	1.000	0.463 (0.152, 1.408)	0.417 (0.162, 1.070)	0.201 (0.065, 0.624) ***	0.0073	0.580 (0.401, 0.841) ^†††^

^1^ Reference category. ^2^ Odds ratio (95% CI). ^3^ *p*-value for linear trend calculated using the median from each quartile and treating it as a continuous variable. ^4^ SD: standard deviation.; * *p* < 0.05, ** *p* < 0.01, *** *p* < 0.005. ^†^ *p* < 0.05, ^†††^ *p* < 0.005; Model 1: Unadjusted. Model 2: Model 1, age, sex, race. Model 3: Model 2, employment, education (≤high school/GED, >high school/GED), smoking (active smoker, not active smoker), total energy intake, MET-minutes/week; Model 4: Model 3, BMI, diabetes, hypertension.

**Table 4 nutrients-17-01636-t004:** Odds Ratio (95% CI) of Low Cognition Across Dietary Quartiles (N = 1053).

	Q1 ^1^	Q2 ^2^	Q3 ^2^	Q4 ^2^	P_Trend_ ^3^	Per SD Increase ^4^
**Alternate Healthy Eating Index 2010**						
**Low Cognition**						
# Cases/Total	60/263	49/263	68/264	42/263		SD = 9.966
Model 1	1.000	0.775 (0.507, 1.183)	1.174 (0.788, 1.749)	0.643 (0.415, 0.997) *	0.2105	0.867 (0.745, 1.008)
Model 2	1.000	0.675 (0.434, 1.051)	0.984 (0.647, 1.496)	0.514 (0.323, 0.818) **	0.0316	0.791 (0.672, 0.931) ^†††^
Model 3	1.000	0.596 (0.291, 1.220)	0.763 (0.366, 1.591)	0.332 (0.130, 0.846) *	0.0443	0.729 (0.542, 0.979) ^†^
Model 4	1.000	0.592 (0.288, 1.219)	0.744 (0.355, 1.559)	0.315 (0.123, 0.807) *	0.0344	0.714 (0.530, 0.962) ^†^
**Healthy Eating Index 2015**						
**Low Cognition**						
# Cases/Total	63/263	53/263	53/264	50/263		SD = 9.192
Model 1	1.000	0.801 (0.530, 1.211)	0.797 (0.527, 1.206)	0.745 (0.490, 1.132)	0.1789	0.928 (0.799, 1.078)
Model 2	1.000	0.813 (0.529, 1.250)	0.700 (0.453, 1.080)	0.685 (0.439, 1.069)	0.0739	0.871 (0.742, 1.022)
Model 3	1.000	0.596 (0.296, 1.202)	0.463 (0.205, 1.043)	0.558 (0.245, 1.268)	0.0970	0.789 (0.590, 1.056)
Model 4	1.000	0.587 (0.290, 1.190)	0.456 (0.201, 1.032)	0.527 (0.229, 1.213)	0.0778	0.773 (0.575, 1.038)
**Alternate Mediterranean Diet**						
**Low Cognition**						
# Cases/Total	65/264	44/202	68/375	42/212		SD = 1.778
Model 1	1.000	0.853 (0.551, 1.318)	0.678 (0.462, 0.995) *	0.756 (0.488, 1.173)	0.1486	0.881 (0.758, 1.024)
Model 2	1.000	0.784 (0.497, 1.235)	0.585 (0.391, 0.875) **	0.600 (0.377, 0.956) *	0.0194	0.810 (0.690, 0.951) ^††^
Model 3	1.000	0.519 (0.213, 1.263)	0.535 (0.248, 1.154)	0.330 (0.129, 0.844) *	0.0346	0.676 (0.495, 0.923) ^†^
Model 4	1.000	0.543 (0.220, 1.338)	0.529 (0.243, 1.149)	0.321 (0.124, 0.830) *	0.0284	0.667 (0.487, 0.914) ^†^

^1^ Reference category; ^2^ Odds ratio (95% CI); ^3^ *p*-value for linear trend calculated using the median from each quartile and treating it as a continuous variable; ^4^ SD: standard deviation; * *p* < 0.05, ** *p* < 0.01; ^†^
*p* < 0.05, ^††^
*p* < 0.01, ^†††^
*p* < 0.005; Model 1: Unadjusted. Model 2: Model 1, age, sex, race. Model 3: Model 2, employment, education (≤high school/GED, >high school/GED), smoking (active smoker, not active smoker), total energy intake, MET-minutes/week. Model 4: Model 3, BMI, diabetes, hypertension.

## Data Availability

Due to participant confidentiality and privacy concerns, data cannot be shared publicly, and requests to access Bogalusa Heart Study data must be submitted in writing. Further information, including the procedures to obtain and access data, is described at https://bogalusaheartstudy.org/research-from-bogalusa-heart-study/ (accessed on 31 March 2025).

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
