# Peer review of "Association Between Dietary Patterns and Cognitive Function in Midlife Adults: The Bogalusa Heart Study"

_nutrients, 2025, doi:10.3390/nu17101636_

Round 1

Reviewer 1 Report

Comments and Suggestions for Authors

Interesting idea of ​​this study, my recommendations are the following:
I recommend mentioning the keywords according to the editing rules, after the abstract.
Introduction – I recommend expanding by mentioning the main objectives and content of the Bogalusa Heart Study. I also recommend expanding the Introduction section by presenting the most relevant aspects regarding the quality of diets. I recommend expanding the background aspects regarding the effects of types of diets in association with cognitive functions in middle-aged people. You present Alzheimer's disease as a disease, but which is specific to elderly people, without referring to the main cognitive diseases in middle-aged people. I recommend expanding and clarifying.
I recommend expanding the aspects regarding socio-economic status and the main vices, such as smoking, in middle-aged people, in relation to the types of diets and cognitive functions.
Line 86 I recommend mentioning the maximum age of the participants targeted in the study, taking into account the average age of adults. Because you only present a minimum of 35 years and over. I recommend mentioning whether informed consent was obtained from the participants to participate in the study as volunteers.
I recommend that the Methods section be divided and numbered according to subsections.
I recommend mentioning the method of collecting the results of this study, in a new section called Procedures.
Lines 138-139 recommend clarifications. Also for all tests and evaluation instruments applied, I recommend expanding the mentioned aspects.
Results – according to the results in the tables, it is observed that participants have a BMI over 30, that is, overweight, in this idea I recommend that this characteristic be presented as well in the inclusion criterion.
Line 289, additional fig. 1 is not found in the attached document, I recommend clarifications.
I recommend that the conclusions of this study be mentioned in a distinct section, called Conclusions.
I recommend that the bibliography be mentioned according to editing rules.

Reviewer 2 Report

Comments and Suggestions for Authors

Thank you for allowing me to review the article titled “ nutrients-3611364_Association between Diet Quality and Cognitive Function in Midlife Adults: the Bogalusa Heart Study”, submitted to the “ Nutrition and Neuro Sciences” section of the journal “Nutrients ”. Below is a critical review with suggestions for improvement.

  1. Title and Fit for Journal Section:

The title appropriately reflects the study’s content, though it could be slightly more specific to highlight the relationship between dietary patterns and cognitive performance. A clearer title may attract the journal's audience more effectively. Additionally, while the manuscript is well-suited to the journal section, a more explicit justification for its relevance would strengthen the introduction.

  1. Abstract:

The abstract is well-structured and summarizes the study effectively. However, expanding on the key findings and their implications for public health and neurodegenerative disease prevention, especially in vulnerable populations, would enhance the clarity and impact.

  1. Introduction:

The introduction establishes the significance of the research area but could better connect the increasing prevalence of Alzheimer’s disease to the potential for dietary interventions. A brief discussion of the inconsistencies in previous research would help justify the current study’s need.

  1. Methodology:

The study design is appropriate, but the use of Odds Ratios (OR) in a cross-sectional study is less suitable than Prevalence Ratios (PR), which would be more accurate in this context. Providing more details on participant inclusion and exclusion criteria would help contextualize the study population.

  1. Results:

Results are clearly presented, but including the specific values for each dietary pattern quartile in the tables and figures would improve clarity and help readers interpret the magnitude of the observed effects more easily.

  1. Discussion:

The discussion effectively places the results within existing research, but further comparison with contradictory studies would be beneficial. A more detailed explanation of the limitations due to the cross-sectional design and potential causal inference would add depth. Suggestions for future research methodologies, such as longitudinal studies, could also strengthen this section.

  1. Conclusion:

The conclusion summarises the findings well, but reinforcing the practical implications of these results for public health interventions would add value. Clearer suggestions for future research directions, particularly regarding longitudinal studies and diverse populations, would improve the conclusion.

  1. References:

The references are appropriate but could benefit from the inclusion of more recent studies, especially those focusing on specific dietary components like omega-3 fatty acids and whole grains.

Comments:

The article addresses an important topic, contributing valuable insights into the relationship between diet and cognitive performance. However, improvements in methodology (use of PR instead of OR), clearer presentation of results, and a more thorough discussion of previous studies and future research directions would enhance the manuscript’s quality and relevance.

Comments on the Quality of English Language

No comments.

Round 2

Reviewer 1 Report

Comments and Suggestions for Authors

No comments